# Methanolic Extract of Phoenix Dactylifera Confers Protection against Experimental Diabetic Cardiomyopathy through Modulation of Glucolipid Metabolism and Cardiac Remodeling

**DOI:** 10.3390/cells13141196

**Published:** 2024-07-15

**Authors:** Laaraib Nawaz, David J. Grieve, Humaira Muzaffar, Arslan Iftikhar, Haseeb Anwar

**Affiliations:** 1Health Biology Laboratory, Department of Physiology, Faculty of Life Sciences, Government College University, Faisalabad 38000, Punjab, Pakistan; laaraibnawaz@gcuf.edu.pk (L.N.); drhumairamuzaffar@gcuf.edu.pk (H.M.); arslaniftikhar@gcuf.edu.pk (A.I.); 2Wellcome-Wolfson Institute for Experimental Medicine, Queen’s University, Belfast BT9 7BL, UK; d.grieve@qub.ac.uk

**Keywords:** diabetic cardiomyopathy, heart failure, *Phoenix dactylifera*, cardio-protection

## Abstract

The incidence of cardiovascular disorders is continuously rising, and there are no effective drugs to treat diabetes-associated heart failure. Thus, there is an urgent need to explore alternate approaches, including natural plant extracts, which have been successfully exploited for therapeutic purposes. The current study aimed to explore the cardioprotective potential of *Phoenix dactylifera* (PD) extract in experimental diabetic cardiomyopathy (DCM). Following in vitro phytochemical analyses, *Wistar albino* rats (N = 16, male; age 2–3 weeks) were fed with a high-fat or standard diet prior to injection of streptozotocin (35 mg/kg i.p.) after 2 months and separation into the following four treatment groups: healthy control, DCM control, DCM metformin (200 mg/kg/day, as the reference control), and DCM PD treatment (5 mg/kg/day). After 25 days, glucolipid and myocardial blood and serum markers were assessed along with histopathology and gene expression of both heart and pancreatic tissues. The PD treatment improved glucolipid balance (FBG 110 ± 5.5 mg/dL; insulin 17 ± 3.4 ng/mL; total cholesterol 75 ± 8.5 mg/dL) and oxidative stress (TOS 50 ± 7.8 H_2_O_2_equiv./L) in the DCM rats, which was associated with preserved structural integrity of both the pancreas and heart compared to the DCM control (FBG 301 ± 10 mg/dL; insulin 27 ± 3.4 ng/mL; total cholesterol 126 ± 10 mg/dL; TOS 165 ± 12 H_2_O_2_equiv./L). Gene expression analyses revealed that PD treatment upregulated the expression of insulin signaling genes in pancreatic tissue (*INS-I* 1.69 ± 0.02; *INS-II* 1.3 ± 0.02) and downregulated profibrotic gene expression in ventricular tissue (*TGF-β* 1.49 ± 0.04) compared to the DCM control (*INS-I* 0.6 ± 0.02; *INS-II* 0.49 ± 0.03; *TGF-β* 5.7 ± 0.34). Taken together, these data indicate that *Phoenix dactylifera* may offer cardioprotection in DCM by regulating glucolipid balance and metabolic signaling.

## 1. Introduction

Diabetic cardiomyopathy (DCM) is defined as ventricular dysfunction in the absence of hypertension or coronary artery disease [1], and it is one of the most common cardiovascular complications, affecting 30–40% of diabetic patients [2]. DCM is characterized by pathological systolic and diastolic dysfunctions, ventricular dilatation, interstitial fibrosis, and cardiomyocyte hypertrophy [3]. Among these, diastolic dysfunction is considered the most prominent feature of DCM [4], and is estimated to be present in 40–60% of diabetic individuals without any significant coronary artery involvement [5].

Persistent hyperglycemia induces alterations in cardiac structure and function, typified by fibrosis, myocardial inflammation and free radical generation via activation of specific signal transduction pathways [6]. Multiple factors including glucotoxicity, lipotoxicity, epigenetic alterations, disrupted calcium signaling, and mitochondrial dysfunction promote adverse cardiomyocyte hypertrophy and extracellular matrix remodeling, leading to systolic and diastolic dysfunction [7]. Passive stiffness of the heart is preferentially increased secondary to metabolic derangement and structural alterations, resulting in abnormal myocardial relaxation [8]. Despite its established incidence and pathological progression, DCM remains largely undiagnosed and may only become evident upon presentation of symptoms [9]. It is therefore critical to identify structural and serological biomarkers to support reliable detection and timely diagnosis of DCM to support improved prevention and progression monitoring.

Despite effective glycemic control using established pharmacological and lifestyle therapies, the incidence of cardiovascular disease in diabetes continues to escalate [10], and there are no specific evidence-based treatments for DCM. Furthermore, because of the adverse side effects of some available drugs, there is an urgent need to explore natural therapeutic alternatives to ameliorate complications associated with diabetes and support improved patient management and prognosis [11]. In this context, there is significant research interest in medicinal plants, which confer therapeutic effects, both in modern medicine and allopathy, due to their pronounced antioxidant and antihyperglycemic properties, reduced side effects, and low cost. Indeed, further to experimental studies highlighting the potential of natural extracts for the management of hyperglycemia, identification of such compounds that can selectively inhibit DCM progression via imposing antihyperglycemic and cardioprotective effects would be of great therapeutic interest.

In this regard, dates (botanical name: *Phoenix dactylifera*), which belong to the *Arecaceae* family of palm and are abundantly cultivated in Middle East countries and North Africa [12], hold exciting promise. There are more than 200 types of dates [13], the largest producer of which is Egypt, while Pakistan, which has the highest global rate of diabetes (26.7%) among its adult population [14], is also a major producer [15]. Dates are enriched in phenolics, flavonoids, procyanidins, carotenoids, and phytosterols [16], which are likely to underlie the reported therapeutic benefits [17], including antitumor, antioxidant, hepatoprotective, antiatherogenic, neuroprotective [18], antihyperlipidemic, nephroprotective, antimicrobial, and gastrointestinal protection [19]. We therefore investigated the potential impact of *Phoenix dactylifera* in experimental DCM, with specific analysis of the functional, serological, histological, and gene expression endpoints, to assess whether dates may hold therapeutic significance for this devastating condition.

## 2. Materials and Methods

### 2.1. Extract Preparation of Phoenix dactylifera L.

*Phoenix dactyleifera* was purchased from a commercial supplier (Faisalabad, Pakistan). The edible portion of the date fruit pulp was isolated manually prior to shade drying and crushing to crude powder using a pestle and mortar. The coarse powder was soaked in methanol at a 1:3 (*w*/*v*) ratio and kept for 48 h at 25 °C in the shade. The solution was then passed through Whatman No. 1 filter paper, as previously described [20], and the filtrate subjected to rotary evaporation (SCILOGEX RE100-Pro; Rocky Hill, CT, USA) at 40 °C until concentrated, and stored for experiments [21].

#### 2.1.1. Qualitative Estimation of Bioactive Phytochemicals

Qualitative analysis for verification of absence or presence of potential phytochemicals, including carbohydrates, proteins, alkaloids, flavonoids, phenols/tannins, and steroids/terpenoids in methanolic extract of *Phoenix dactylifera* was performed using standard procedures [22].

#### 2.1.2. Quantitative Estimation of Bioactive Phytochemicals

##### Total Phenolic Constituents (mg of Gallic Acid Equivalent/g Dry Weight of Plant)

To quantify the total phenolic content (TPC) in methanolic extract of date fruit, 30 μL of methanolic extracts was diluted with an equal volume of Folin Ciocalteu reagent (1 mg/mL) and 600 μL of 2.5% Na_2_CO_3_. The amalgam was left at room temperature for 1 h prior to measurement of the optical density (OD) at a wavelength of 760 nm using a biochemistry analyzer (Biolab-310; Biobase, Jinan, China). A gallic acid standard curve (0.789 to 200 μg/mL) was utilized for evaluation of the TPC [23].

##### Total Flavonoid Content (mg of Quercetin Equivalent/g Dry Weight of Plant)

To analyze the total flavonoid content, date fruit extract (100 μL) was diluted in 1 mL distilled water. Following incubation for 5 min at room temperature (RT), 125 μL AlCl_3_ and 75 μL of 5% NaNO_2_ were added and placed at RT for a further 6 min. Afterwards, 125 μL of 1 M NaOH and 2.5 mL distilled water were added and absorbance measured at 540 nm using a biochemistry analyzer (Biolab-310) with reference to quercetin as the standard control (0 to 100 μg/mL) [23].

#### 2.1.3. Determination of Antihyperglycemic Activity by α-Amylase Inhibition Assay

α-Amylase inhibitory activity was evaluated, as previously described [24], with minor modifications. The formula below was used to calculate the % enzyme inhibition.
(1)$$\alpha -amylase\;inhibition\left(\%\right)=\frac{A\;blank-A\;sample}{A\;blank}\times 100$$

#### 2.1.4. Identification and Quantification of Phenolic Constituents by HPLC

High-performance liquid chromatography (HPLC) was used to identify and quantify the phenolic content. Briefly, date extract was liquefied in methanol (250 μg/mL) and, subsequently, subjected to liquid chromatography on the HPLC system (Perkin Elmer^®^, Washington, DC, USA), linked with a Flexer Binary LC pump, a UV/VIS LC Detector (Shelton CT^®^, Shelton, CT, USA), and a reverse-phase C18 column (5 mm, 250 × 4.6 mm). Data were analyzed using Chromera software (version 4.1.2.6410) [25].

### 2.2. In Vivo Experimental Trial

#### 2.2.1. Animal Housing and Induction of Diabetes

Albino Wistar rats (N = 16) weighing 60 ± 10 g (age: 2 ± 1 weeks) were bred and maintained at the Department of Physiology, Government College University, Faisalabad, Pakistan. It has been documented that early administration of a high-energy diet effectively induces metabolic syndrome [26,27]. Rats were housed under standard conditions (Figure 1) (40–60% ambient humidity, 12 h light/dark cycle, and 26 ± 2 °C) and fed either a normal or high-fat diet (HFD; 30%) with 5% sucrose in drinking water ad libitum throughout the experiment [27]. To induce type 2 diabetes mellitus (T2DM), HFD rats were administered low-dose streptozotocin (STZ, 35 mg/kg i.p.) after two months to promote pancreatic β-cell destruction and insulin resistance [27,28]. This model effectively induces hyperglycemia, insulin resistance, dyslipidemia, which is reflective of clinical T2DM as the basis for study of innovative intervention strategies [29]. Fasting blood glucose (FBG) levels were monitored to confirm hyperglycemia (FBG ≥ 150 mg/dL at 72 h) using a glucometer (OnCall^®^ Ez-II; SN 303S0014E09, South Croydon, UK) and rats assigned to 4 experimental DCM groups (N = 4): (1) healthy control, normal chow diet, and drinking water ad libitum; (2) DCM control, HFD, and 5% glucose in drinking water ad libitum; and (3) DCM metformin, as group 2 + metformin 200 mg/kg/day orally via gavage; DCM PD, as group 2 + methanolic extract of *Phoenix dactylifera* 5 mg/kg/day orally [30].

Fasting blood glucose (FBG) was monitored weekly. After 25 days of treatment, rats were sacrificed by cervical dislocation. Blood was collected and centrifuged at 2000× *g* for 10 min and serum kept at −20 °C until further investigation. For histopathology, heart and pancreatic tissue samples were excised and preserved in 10% formalin. The relative organ weight of heart was measured using the following formula [31]:(2)$$Relative\;organ\;weight=\frac{\left(Organ\;weight\right)}{Bodyweight}\times 100\;$$

#### 2.2.2. Biochemical Parameters

##### Determination of Glucolipid Profile

Serum glucose was quantified using a commercially available diagnostic SBio glucose kit (Ref: 90504250; Biosciences, Yishun, Singapore). Serum insulin was quantified by ELISA (Calbiotech; Catalog No. IN374S; El Cajon, CA, USA). HbA1c was measured with HbA1c OSR-6192 Beckman Coulter^®^ DxC 700 AU [32]. The Crescent Diagnostic (Jeddah, Saudi Arabia) reagent kit method was used to measure total cholesterol, triglycerides, and high-density lipoprotein–cholesterol spectrophotometrically (BIOLAB-310). The concentration of low-density lipoprotein–cholesterol was calculated by the following equation:(3)$$LDL-Cholesterol=\left(Total\;Cholesterol\right)-\left(\frac{Triglycerides}{5}\right)-\left(HDL-Cholesterol\right)$$

For determination of the cardiac risk ratio (CRR) and cardioprotective index (CPI), concentrations (mg/dL) of the total cholesterol, HDL-c and LDL-c were converted into mmol/L and calculated using the following formulae [33]:(4)Cardiac risk ratio = Total cholesterol/HDL
(5)$$Cardioprotective\;index=\frac{HDL}{LDL}$$

##### Determination of Myocardial Profile

Levels of pro-BNP were analyzed using immunoassay kits by Elecsys proBNP II Ref (04842464-190) cobas^®^; Roche, Mannheim, Germany [34]. Electrocardiograms (ECG) were recorded using PowerLab (ADInstruments data acquisition system; Colorado Springs, CO, USA) [35]. Electrolytes (sodium and potassium) were analyzed using Electrolytes Fully automated AU 700 (Medica corporation; Beckman Coulter; Danvers, MA, USA). The levels of creatine kinase (CKMB) and lactate dehydrogenase (LDH) were estimated spectrophotometrically (BIOLAB-310; Biosciences, Yishun, Singapore) according to the manufacturer’s specifications [36]. AST was determined by adapting the standard procedures of LAB KIT (REF-30243; Chemeflex, Canovelles Barcelona) for AST. The markers of oxidative stress were (total antioxidant capacity (TAC), total oxidant status (TOS), and malondialdehyde (MDA)) were analyzed spectrophotometrically (BIOLAB-310; Biosciences, Yishun, Singapore), as previously described in [37,38], with minor modifications.

#### 2.2.3. Tissue Analyses: Gene Expression and Histopathology

The RT-qPCR method was used to detect mRNA expression levels in pancreatic and heart tissue. Total RNA was extracted using TRIzol reagent (Invitrogen, Waltham, MA, USA) and evaluated for concentration and purity using Nanodrop 2000 spectrophotometer (Thermo Fisher Scientific, Waltham, MA, USA). Total isolated mRNA was reverse transcribed to cDNA using the RevertAid cDNA synthesis kit (Thermo Fisher Scientific, Waltham, MA, USA) according to the manufacturer’s manual. Quantitative RT-qPCR was carried out on the iQ5 Bio-RAD machine using Maxima SYBR Green/ROX qRT-PCR Master Mix (Thermo Fisher Scientific, Waltham, MA, USA). Expression profiles of *Ins-1*, *Ins-2*, *Pdx1*, *MafA*, and *Glut-2* were quantified in the pancreas, while *NF-kB*, *TGF-β*, and *TNF-α* were quantified in the heart, with normalization to β-actin and *GAPDH* as housekeeping/reference genes. Specific primer sequences were used to amplify the genes (Table 1). For the histopathology, pancreatic and heart tissues (left ventricle) were recovered from formalin (10%) and paraffinized to construct blocks. For analysis, 4–5 μm sections were sliced using a microtome (Bk-Mt268m; Biobase Biodustry (Jinan, China) Co., Ltd.) and fixed on an albumin-coated glass slide and stained with hematoxylin and eosin prior to examination under a light microscope at 40× magnification.

#### 2.2.4. Effect of P-Coumaric Acid on In Vitro Cell Culture of Human Cardiac Fibroblasts

To investigate the antifibrotic efficacy of p-coumaric acid (PCA; compound detected in HPLC) under hyperglycemic conditions, normal human cardiac fibroblasts (HCFs; source: ScienCell™ product #6330; Carlsbad, CA, USA) were stimulated with hyperglycemia, together with *TGF-β*, to induce a profibrotic response. For this purpose, HCF cells were cultured in Dulbecco’s modified eagle medium (DMEM) supplemented with 10% fetal bovine serum (FBS) at 37 °C in atmosphere containing 5% CO_2_ up to passage 13. The cell viability was determined using an MTT assay in the presence or absence of PCA. Briefly, cells were incubated with varying concentrations of PCA (15–45 μΜ); cells without the drug were used as a control. Thiazolyl blue (MTT) was dissolved in PBS and incubated with the cells for 3 h, before washing with PBS and the addition of DMSO. After 15 min, the absorbance was measured at 570 nm using a microplate reader. After determining the desired dosage of PCA, experiments to investigate antifibrotic efficacy of PCA under hyperglycemic conditions were initiated. The cells were seeded at a density of 0.3 × 10^6^ into 6-well plates under standard conditions. When the cells reached a 70–80% confluency, the spent medium was aspirated and replaced with serum-free medium (serum starvation for 24 h) before assigning to the following three groups:i.Vehicle control (VC): DMEM containing 0.1% DMSO;ii.Induction (TGF-*β* + D): stimulation with TGF-*β* (10 ng/mL) in the presence of hyperglycemia (25 mmol/L D-glucose) for 48 h;iii.Treatment (PCA): Stimulation with TGF-β in the presence of hyperglycemia (25 mmol/L D-glucose) followed by treatment with PCA (30 μΜ; 48 h). PCA was dissolved in DMSO prior to dilution in DMEM, with a final concentration in experiments not exceeding 0.1% (*v*/*v*).

After the treatments, cells were collected and stored at −80 °C for further analyses. For the Western blot analysis, HCF cells were lysed in RIPA buffer, including proteinase inhibitor cocktail and phosphatase inhibitor. Total lysates were vortexed and incubated on ice for 15 min prior to centrifugation at 13,300 RCF for 20 min at 4 °C. Supernatants were then collected and protein concentration estimated using a bicinchoninic acid assay kit (Thermo Scientific, Waltham, MA, USA). Western blot was performed by lysing the sample with standard sample buffer. After boiling the samples for 5 min, protein samples (10 μg) were fractionated by 10% SDS–PAGE before transfer to PVDF membrane and blocking with 5% nonfat milk for 1 h. The membrane was then probed with primary antibody against α-SMA (mouse monoclonal anti-α smooth muscle actin, SMA 42 kDa, 1∶1000, Sigma-Merck; Burlington, MA, USA) and incubated at 4 °C on a rotating device overnight. Following a sequence of washes in TBST, the membrane was incubated with secondary antibody for 1 h and the labeled protein of interest exposed to X-ray in a dark chamber using enhanced chemiluminescence (ECL kit). Western blot bands were quantified using ImageJ software (ImageJ 1.54d, Java 1.8.0_345) and normalized to β-actin as the reference control. The membrane was then stripped, re-blocked, and probed with anti-CTGF antibody (CTGF 38 kDa; 1:1000; ab209780) overnight, before washing with TBST and probing with secondary antibody for 1 h, and further washing with TBST and imaging in a G-box chamber. For the gene expression analysis, RNA was extracted from cell pellets following the manufacturer’s protocol (Roche; High Pure RNA Isolation Kit; Cat# 11-828-665-001), quantified (Thermo Scientific Nanodrop One^c^), and subjected to cDNA synthesis (Thermo Scientific RevertAid First Strand cDNA synthesis kit). Specific primers (Table 1) for *CTGF* and *α-SMA* were used for the qRT-PCR gene expression analysis with reference to β-actin as the housekeeping gene.

#### 2.2.5. Statistical Analysis

Data are represented as the mean ± standard error of the mean (SEM). Unpaired Student’s *t*-test was used to assess the statistical significance between two groups, and one-way analysis of variance (ANOVA) with Bonferroni post hoc testing was used to compare multiple groups [39], using GraphPad Prism 8.0.263 software (San Diego, CA, USA). *p* < 0.05 was taken to indicate statistical significance.

## 3. Results

### 3.1. In Vitro

#### 3.1.1. Qualitative and Quantitative Analysis of *Phoenix dactylifera*

Qualitative analysis of the methanolic extract of *Phoenix dactylifera* L. (PD) (Table 2) detected various phytochemicals, including phenols, alkaloids, anthraquinones, flavonoids, reducing sugar, saponins, terpenoids, and steroids. Quantitative estimation of the total flavonoid content (TFC) and total phenolic content (TPC) further indicated that PD methanolic extract is enriched with flavonoids and phenols.

#### 3.1.2. Quantitative Estimation of Antihyperglycemic Activity and HPLC

The α-amylase inhibition assay confirmed the strong antihyperglycemic efficacy of the PD extract when tested against acarbose (Acr) as the standard drug control. The PD extract and Acr showed similar antihyperglycemic activity and concentration-dependent increase in antihyperglycemic efficacy (Figure 2A,B). Moreover, the HPLC analysis confirmed the detection of p-coumaric acid (phenolic acid) as the active bioconstituent of PD methanolic extract, present at high concentrations as signified by the indicated peak on the representative trace (Figure 2C).

### 3.2. In Vivo

#### 3.2.1. *Phoenix dactylifera* Modulates Glucolipid Metabolic Alterations in DCM

Effective glycemic control is the cornerstone of managing T2DM, as it can reduce the risk of microvascular complications associated with chronic hyperglycemia. The levels of HbA1c, serum glucose, insulin, and fasting blood glucose were significantly elevated in the DCM controls (HbA1c 9.8 ± 0.61; glucose 199.5 ± 9.8 mg/dL; insulin 27 ± 3.41 ng/mL) versus healthy controls, while these changes were normalized by PD treatment (HbA1c 5.1 ± 0.23; glucose 82.5 ± 4.5 mg/dL; insulin 17 ± 2.45 ng/mL) to an equivalent level to metformin treatment (Figure 3A–D). Furthermore, PD extract significantly lowered levels of total cholesterol (TC), triglycerides (Tg), and low-density lipoproteins (LDLs) versus DCM control, whilst restoring reduced HDL levels (Figure 3). Metabolic responses to metformin and PD extract in DCM animals were broadly similar indicating therapeutic potential for glucolipid control offered by PD extract.

##### *Phoenix dactylifera* Upregulates Transcriptional Factors and Genes Involved in Insulin Signaling Pathway and Preserves Pancreatic Structural Integrity in DCM

Insulin signaling pathway genes and transcriptional factors are known to have a significant role in regeneration, proliferation and secretion of insulin from pancreatic β-cells. Th results (Figure 4A–E) show that the levels of *INS-I*, *INS-II*, *PDX-1*, *MAFA*, and *GLUT-2* were downregulated in the DCM control (*INS-I* 0.65 ± 0.02; *INS-II* 0.49 ± 0.02; *PDX-1* 0.78 ± 0.03; *MAFA* 0.43 ± 0.02; *GLUT-2* 0.25 ± 0.03) versus healthy control, whilst both treatment groups, DCM metformin and DCM PD (*Phoenix dactylifera)*, showed upregulated expression levels of *INS-I* 1.69 ± 0.02; *INS-II* 1.3 ± 0.02; *PDX-1* 1.45 ± 0.02; *MAFA* 1.1 ± 0.01, and *GLUT-2* 1.16 ± 0.02, which is consistent with improved insulin functioning and glycemic control (Figure 3). Moreover, histopathological analyses of pancreatic tissue affirmed that the structural integrity of islets of Langerhans was preserved and that both PD and metformin afforded protection against high-fat diet and streptozotocin-induced pancreatic damage (Figure 4F).

#### 3.2.2. *Phoenix dactylifera* Reduces Myocardial Oxidative Stress and Protects against Injury

Myocardial levels of pro-BNP were significantly elevated in the DCM controls versus the healthy controls (healthy control: 98 ± 4 pg/mL; DCM control: 420 ± 3 pg/mL) with suppressed induction evident in both treatment groups (DCM metformin: 220 ± 4; DCM PD: 218 ± 3 pg/mL; Figure 5A). These changes were associated with restoration of heart rate (185 ± 4 BPM) and QRS complex duration (0.078 ± 0.01 ms; Figure 5B,C) in the DCM controls. The levels of myocardial enzymes were also significantly increased in the DCM controls (LDH 2095 ± 53.5 IU/L, CKMB 2185 ± 35.24 IU/L, AST 599.5 ± 24.84 IU/L) with concomitant decrease in total antioxidant capacity (TAC; 1.1 ± 0.23 mmol Trolox equiv./L) and increase in oxidative stress markers (TOS 171 ± 10.21 µmol H_2_O_2_equiv./L; MDA 0.46 ± 0.09 mmol/L) versus the healthy controls (LDH 1102 ± 19 IU/L; CKMB 982 ± 12 IU/L; AST 182 ± 11 IU/L; TAC 2.5 ± 0.25 mmol Trolox equiv./L; TOS 48 ± 12 µmol H_2_O_2_equiv./L; MDA 0.17 ± 0.07 mmol/L) (Figure 5D–I). Interestingly, all of these changes were positively impacted by the PD extract, which showed superior effects to metformin in relation to levels of myocardial enzymes, TAC, and TOS and shortening of the QRS complex. Taken together, these data highlight effective cardioprotection by PD extract against experimental DCM, over and above that conferred by metformin at equivalent glucolipid profiles, suggesting that PD may mediate direct actions on the myocardium.

#### 3.2.3. *Phoenix dactylifera* Confers Cardioprotection against DCM by Downregulating Pro-Inflammatory and Pro-Fibrotic Genes’ Expressions

Myocardial expression levels of pro-inflammatory genes (*NF-ĸB* and *TNF-α*) and the profibrotic gene, *TGF-β*, were upregulated in DCM controls (*NF-ĸB* 3.14 ± 0.03; *TNF-α* 4.81 ± 0.01; *TGF-β* 5.67 ± 0.04; Figure 6A–C) versus healthy controls, indicative of hyperglycemia-induced low-grade inflammation and fibrosis, and associated with deranged tissue structure, characterized by inter-fiber edema and prominent fibrosis (Figure 6D). Treatment with metformin significantly but incompletely restored gene expression changes and myocardial structure in DCM, whilst PD treatment resulted in virtually equivalent gene expression and myocardial structure compared to healthy controls. In addition, PD extract restored relative organ weight and cardiomyocyte diameter as compared to DCM controls (Figure 6D,E), indicating potential of PD extract in offering protection against inflammation, fibrosis and pathological hypertrophy in DCM. These data add further support to those presented in Figure 4, indicating that PD confers superior myocardial benefit to metformin in experimental DCM by promoting direct protective effects against adverse cardiac remodeling.

#### 3.2.4. p-Coumaric Acid Inhibits Hyperglycemia and TGF-β Stimulated Fibrosis in Human Cardiac Fibroblasts

The potential cytotoxic effects of PCA were investigated by exposure of HCFs to a range of PCA concentrations (15–45 µM) for 3 h. The MTT assay indicated that cell viability remained similar with PCA treatments (Figure 7A); a concentration of 30 µM was selected for use in further experiments. The mRNA expressions of both CTGF and α-SMA were markedly upregulated by combined hyperglycemia and TGF-β stimulation and reduced by ~50% by the PCA treatment (Figure 7B,C), while changes in the protein expressions of CTGF and α-SMA showed a similar pattern (Figure 7D–F). Taken together, these data indicate that p-coumaric acid inhibits myofibroblast differentiation, adding further support to our suggestion that this compound confers a direct cardiac benefit.

## 4. Discussion

T2DM is a complex, polygenic, and heterogenous disorder characterized by insulin resistance and pancreatic β-cell dysfunction, leading to poor glycemic control as a major risk factor correlating with poor metabolic and cardiovascular outcomes, for which current treatments are suboptimal. In this regard, we investigated methanolic extract of *Pheonix dactylifera* (PD) as a potential regulator of glucolipid metabolic and cardiovascular dysfunction in experimental T2DM, both in vivo and in vitro. Glucolipid metabolic profiling (Figure 3) depicted elevated levels of fasting blood glucose (FBG), HbA1C, serum glucose, insulin, total cholesterol, triglycerides, and LDL-c in the DCM control rats compared with healthy controls. Indeed, this model is characterized by established STZ-induced β-cell destruction and insufficient insulin action [40], combined with high-fat diet-mediated insulin resistance and suppression of postprandial fatty acid release as a key driver of elevated triglycerides and lipoprotein dysregulation [41]. Notably, metabolic alterations in circulating lipids pose the highest risk for cardiovascular complications in T2DM [42]. These detrimental changes were largely restored by treatment with both PD and metformin as the comparator drug control which acts to reduce lipid secretion from intestinal epithelial cells by AMPK and GLP-1 (glucagon-like peptide-1) activation [43]. Interestingly, our data suggest that the reduction in glucolipid parameters and increase in HDL-c observed in DCM rats subjected to PD treatment are likely to be due to the antihyperglycemic efficacy of PD extract, as indicated by α-amylase inhibition assay (Figure 2). Previously, it was documented that PD extract mediates its significant antihyperglycemic effects through inhibition of α-amylase and glucosidase [44], and stimulation of endogenous release of insulin from extra-islet sources [45], which could be due to dietary fibers of PD extract which reduce carbohydrate gastrointestinal absorption and enhance glucose uptake by skeletal muscles [46], while the evident antihyperlipidemic effect of PD extract may be mediated through direct or indirect inhibition of endogenous cholesterol synthesis [47]. As a consequence, PD extract reduced the cardiac risk ratio and conferred strong cardioprotection in experimental DCM (Figure 3).

To explore impact of PD extract on pancreatic tissue function and expressions of genes and transcriptional factors central to insulin signaling and maintenance of β-cell regeneration, proliferation, and secretion, a qRT-PCR mRNA analysis was conducted and revealed (Figure 4) reduced expressions of *INS-I*, *INS-II*, *MAFA*, *GLUT-2*, and *PDX-1* in the DCM control rats versus healthy controls, further to specific STZ-induced β-cell destruction [48], which was confirmed by histopathology [49]. As expected, metformin treatment upregulated expression levels of *INS-I*, *INS-II*, *MAFA*, *GLUT-2*, and *PDX-1* in DCM rats which was associated with β-cell restoration, consistent with its established role as both an insulin sensitizer and promoter of insulin secretion [50]. Similar effects on pancreatic gene expression and morphology were observed in the PD-treated DCM rats, which may be attributed to flavonoid-mediated regulation of insulin signaling and secretion, carbohydrate digestion, and tissue glucose uptake [51]. Indeed, the high phenolic and flavonoid contents of the PD extract was confirmed in the current study (Table 2), together with significant α-amylase inhibitory activity (Figure 2) as an indicator of the antihyperglycemic efficacy of PD and capability of supporting normal β-cell structure and functioning. Taken together, these data provide a strong indication that PD extract preserves pancreatic islet insulin signaling in experimental T2DM as the basis for its emerging therapeutic potential.

As uncontrolled or chronically elevated glycemia typically leads to adverse cardiac remodeling, we also investigated the efficacy of PD extract on T2DM rat hearts. As cardiomyocytes have low innate capacity of antioxidants, such as glutathione and superoxide dismutase, the myocardium is comparatively more prone to deleterious effects of free radicals [52]. Indeed, hyperglycemia promoted increased myocardial reactive oxygen species (ROS) generation, as depicted by the high total oxidant status (TOS) and malondialdehyde (MDA) levels in the DCM versus control rats with concomitant increased levels of myocardial enzymes (LDH, CK-MB, and AST; Figure 5). These changes were associated with elevated pro-BNP, as well as decreased heart rate and QRS complex widening, as evidence of myocardial stress and functional deterioration [53]. These aberrant electrophysiological changes could be due to de novo ceramide synthesis, and ceramide induced HERG depression leading to arrythmias and heart failure in DCM. As oxidative stress mediates ceramide’s effects, this, at least, partially explains the role of antioxidants in protecting against aberrant electrophysiological changes [54]. Treatment of DCM rats with either metformin or PD, rescued antioxidant activity whilst reducing myocardial enzyme expression and oxidative stress and improving cardiac function. Interestingly, impact of PD on these indices was superior to that of metformin, despite equivalent metabolic effects, suggesting that PD confers direct cardioprotective benefit via divergent mechanisms. Indeed, metformin is known to indirectly suppress oxidative stress by reducing activity of antioxidants, such as superoxide dismutase, and mitochondrial complex I [55], whereas antioxidant activity of PD extract is mediated via various phenolic compounds which are likely to have reinforced myocardial antioxidative defense in our experimental DCM model leading to additional functional benefit [47]. Previous studies have also indicated that PD extract repressed hyperglycemia induced cardiomyopathy by suppressing inflammation and myocardial enzyme expressions [56], thereby protecting against adverse cardiac remodeling through reduced oxidative stress reflected by decreased MDA and increased superoxide dismutase (SOD) levels [57].

As T2DM is characterized by chronic low-grade inflammation, which triggers the release of various fibrotic mediators and promotion of cardiac fibrosis, the expression levels of proinflammatory genes, TNF-α and NF-κB, and the profibrotic gene, TGF-β, were quantified in myocardial tissue to further investigate apparent cardioprotective effects of PD extract. These targets were selected as the NF-κB pathway activation represents a central convergence point for numerous cytokines and chemokines in T2DM [58], which increases the production of proinflammatory cytokines, including TNF-α [59], resulting in a vicious cycle of chronic low-grade inflammation and upregulation of TGF-β, as a key mediator of cardiovascular fibrosis [60]. In addition, under inflammatory stimulation, abnormal deposition of extra cellular matrix is increased with concomitant increase in collagen fibers, ultimately aggravating myocardial fibrosis [61].

Indeed, mRNA expressions of NF-κB, TNF-α, and TGF-β were markedly increased in the DCM controls in parallel with immune cell infiltration and myocardial histopathology characterized by inter-fiber edema and prominent fibrosis (Figure 6), impacts that were significantly reduced by both interventions but to a greater extent by PD versus metformin treatment. As the superior inhibitory effect of the PD extract on pro-inflammatory and profibrotic signals in myocardial tissue could be due to its high content of polyphenols, which modulate activity of cyclo-oxygenase and lipo-oxygenase activity [62] and downstream NF-κB and TNF-α. Similarly, the more pronounced downregulation of TGF-β and improved myocardial histopathology observed with the PD extract likely occurred secondary to its combined inhibitory actions on inflammation and free radical production, as established drivers of fibrosis [63]. Under basal conditions, very little oxygen is converted to ROS, whereas hyperglycemia promotes excessive ROS generation leading to activations of NF-κB and TGF-β [64]. This implies that downregulated expressions of NF-κB, TNF-α, and TGF-β by the PD extract could also occur secondary to its antioxidant effects.

Myocardium is particularly prone to the deleterious effects of glucotoxicity and lipotoxicity, as both nutrients promote myofibril disorganization and disarray as observed in failing hearts [65]. Notably, treatment with PD extract resulted in a more normalized structure of myofibers in DCM tissue together with less fibrosis compared to the metformin treatment group which continued to exhibit myofiber disarray. This term specifically refers to nonparallel organization of cardiac myocytes [66] which is associated with regions of myocardial scarring [67]; its presence in metformin treated hearts is a further indicator that this established antihyperglycemic drug confers some protection against cardiovascular damage [68], while reduced myofiber disarray seen in PD extract provides evidence of superior cardioprotective efficacy. In addition, the relative heart weight index was significantly higher in the DCM control, suggestive of pathological hypertrophy in response to hyperglycemia and associated oxidative stress as compared to the healthy control and was restored by treatment with both the PD extract and metformin. Such protective effects are likely to be, at least, partly due to the ability of the PD extract to enhance endogenous antioxidant in the myocardium [16], leading to a reduction in injury markers together with edema and myonecrosis and restoration of structure [69]. It should be noted that despite the increased relative heart weight in the DCM controls, the cardiomyocyte diameter significantly increased compared to the healthy control and treatment groups, which could be due to long-standing metabolic derangements leading to cardiomyocyte hypertrophy, atrophy, and degeneration [70].

Although our in vivo data clearly indicate that the PD extract conferred more marked cardioprotection versus metformin, which is likely to be mediated via more pronounced anti-inflammatory and antioxidant effects, these experiments did not allow us to conclude whether these effects may be due to direct actions of PD extract on the heart or entirely mediated by indirect glucometabolic alterations. We therefore conducted complementary in vitro studies to assess the impact of p-coumaric acid, (PCA; bioactive constituent of PD extract detected in HPLC) on cultured human cardiac fibroblasts (HCFs) exposed simultaneously to hyperglycemic and profibrotic stimuli to promote phenotypic change and differentiation to activated myofibroblasts. mRNA and protein expression of myofibroblast markers, α-smooth muscle actin (α-SMA) and connective tissue growth factor (CTGF), which promote extracellular matrix deposition and fibrosis, were markedly increased after 48 h stimulation with high glucose and TGF-β, and significantly reduced by PCA treatment (Figure 7). These data provide additional support to our in vivo findings, indicating that PD extract reduces profibrotic signaling via its bioactive constituent, PCA, whilst also providing evidence that PD extract exerts direct cardiac benefits in DCM in parallel to metabolic improvement. Taken together, these results suggest that methanolic extract of *Phoenix dactylifera* can offer significant cardioprotection by orchestrating adverse cardiac remodeling in experimental DCM by modulating both glucolipid metabolism and fibroblast activation.

## 5. Limitations of the Study

While the presented results clearly support the therapeutic effects of PD extract against experimental DCM, it is important to highlight some limitations that should be considered when interpreting these data. We chose to focus on a single daily oral dose of PD extract based on initial analysis of the glycemic efficacy and published reports. However, it would have been interesting to explore a range of doses and different routes of administration to investigate whether its therapeutic effects could be further enhanced. The duration and point of commencement of PD treatment are also likely important determining factors of therapeutic efficacy. In this study, animals were treated with PD extract two months after the induction of T2DM for a period of 25 days, both of which are relatively acute in the context of DCM as a chronic condition. It would, therefore, be interesting to both extend the treatment duration while also studying the impact of earlier and later interventions to assess whether PD treatment may prevent DCM development or reverse established DCM. In our study, the impact of PD extract on structural and functional cardiac remodeling in experimental DCM was assessed through in vivo analysis of electrocardiograms and assessments of tissue histology and gene expressions. Notably, the observed effects of PD extract on these indices were larger superior to those conferred by metformin, as a comparator drug exerting equivalent glucolipid alterations, whilst PCA inhibited in vitro myofibroblast differentiation, providing clear evidence of direct and specific cardio-protection against adverse remodeling in DCM. Nonetheless, it will be important to incorporate echocardiography analysis into future studies to further define the impact of PD extract on cardiac structural and functional remodeling, including investigation of differential efficacy against systolic and diastolic dysfunctions, toward potential therapeutic application.

## 6. Therapeutic Potential of PD Extract and Future Recommendations

On the basis of our findings, PD extract possesses evident potential to restore glucolipid metabolism, which is dysregulated in DCM, while also protecting the myocardium from adverse cardiac remodeling in response to associated hyperglycemia (Figure 8). These intriguing results provide an important foundation for future studies to define the optimal dose and formulation to achieve maximal efficacy against experimental DCM. Further preclinical studies are required to interrogate detailed impacts of PD extract on cardiac systolic and diastolic functions and structural remodeling, combined with clinical investigation incorporating longer-term treatment, to validate its exciting therapeutic potential.

## 7. Conclusions

*Phoenix dactylifera* extract improved glycemic index, restored glucolipid metabolic alterations, and upregulated insulin signaling pathway genes in the experimental DCM, conferring both direct and indirect cardioprotective effects against adverse cardiac remodeling due to its rich phenolic, flavonoid, and antioxidant contents. As currently available drugs for clinical DCM are nonspecific and somewhat ineffective and may even promote health-related complications, date fruit holds clear potential as an excellent natural and safe therapeutic source to not only protect against diabetes-associated cardiomyopathy but also to fulfill key nutritional requirements.

## Figures and Tables

**Figure 1 cells-13-01196-f001:**
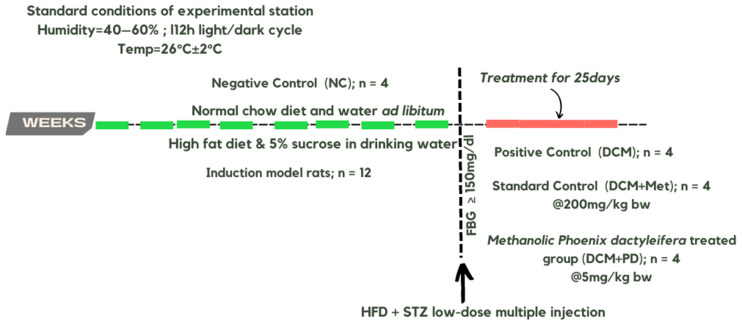
Consort diagram showing the protocol for the in vivo experimental trial.

**Figure 2 cells-13-01196-f002:**
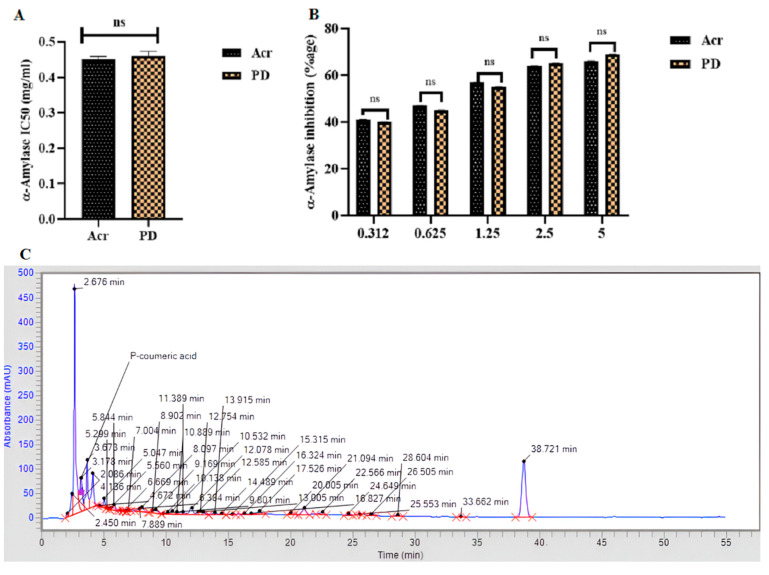
*Phoenix dactylifera* metabolic extract showing an equivalent antihyperglycemic activity to acarbose, which is mediated by p-coumaric acid as its active bio-constituent. α-Amylase inhibition assay with comparison between PD and acarbose (Acr) as the standard drug control: (**A**) quantification of IC50; (**B**) concentration-dependent inhibition of α-amylase inhibition, mean ± SEM, N = 3; (**C**) representative HPLC trace showing detection of p-coumaric acid in PD extract signified by the indicated peak.

**Figure 3 cells-13-01196-f003:**
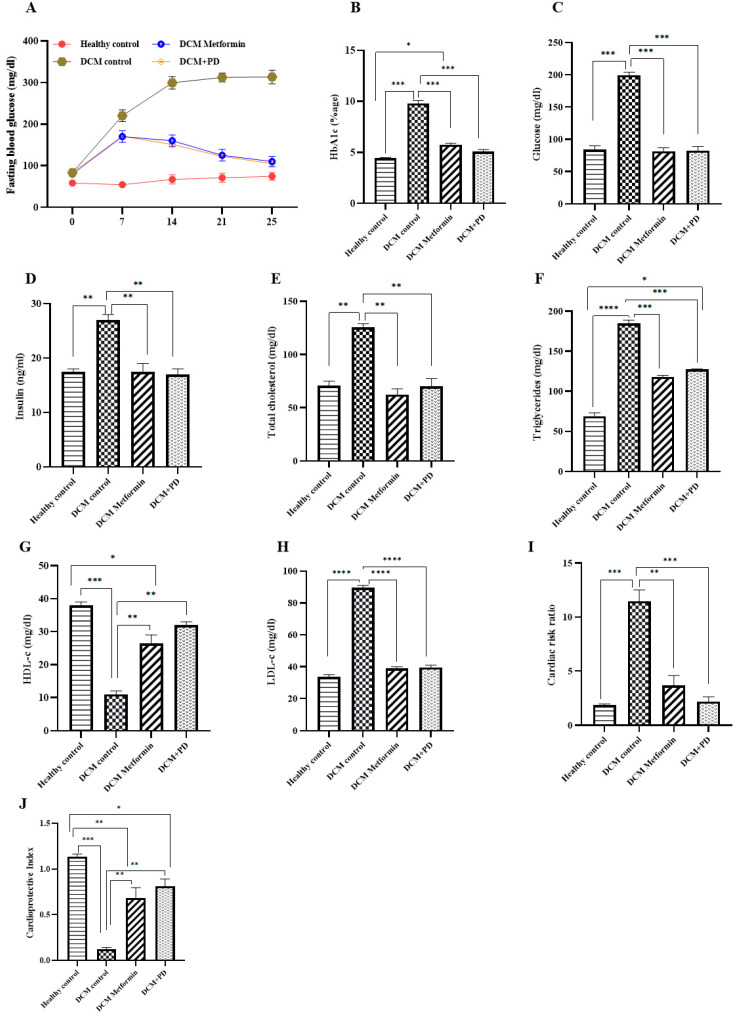
*Phoenix dactylifera* treatment restored glucolipid metabolic dysfunction in DCM: (**A**) progression of fasting blood glucose levels in healthy control and DCM control rats treated with or without PD or metformin. After 25 days, rats were sacrificed for serum analysis of (**B**) glucose, (**C**) HbA1c, (**D**) insulin, (**E**) total cholesterol, (**F**) triglyceride, (**G**) HDL-cholesterol, (**H**) LDL-cholesterol (**I**) cardiac risk ratio, and (**J**) cardioprotective index. Data are Mean ± SEM, N = 4. **** *p* < 0.0001, *** *p* < 0.001, ** *p* < 0.01, * *p* < 0.05.

**Figure 4 cells-13-01196-f004:**
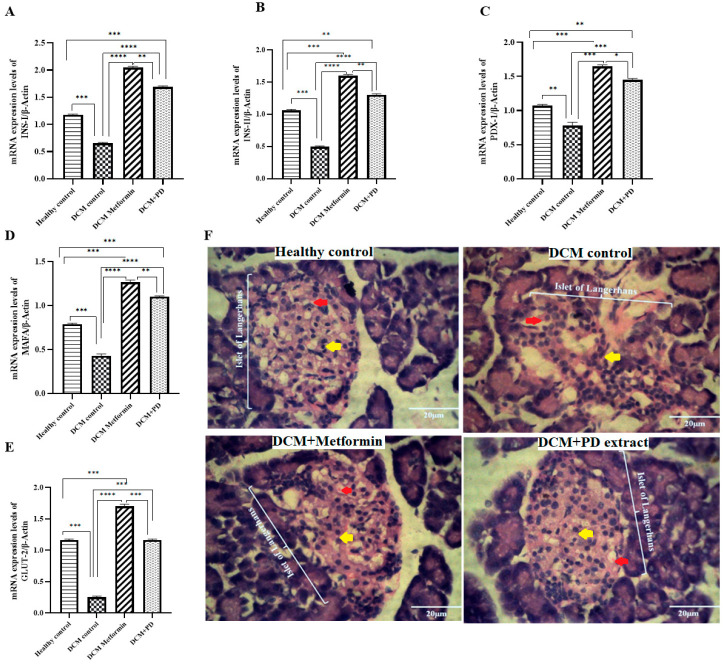
*Phoenix dactylifera* activates insulin signaling pathway genes and transcriptional factors and improves pancreatic islet morphology in DCM. Gene expression of (**A**) *INS-I*, (**B**) *INS-II*, (**C**) *PDX-1*, (**D**) *MAFA*, and (**E**) *GLUT-2* in the pancreas of the healthy control and DCM control rats treated with or without PD or metformin, by RT-qPCR, with normalization to β-actin as housekeeping control. Data are mean ± SEM, N = 4. **** *p* < 0.0001, *** *p* < 0.001, ** *p* < 0.01, * *p* < 0.05. (**F**) Representative H&E-stained histopathology sections (40× magnification) showing normal histological features islets of Langerhans, with scattered β cells (yellow arrows) and red blood cells (red arrows) visible in the vicinity (EX: exocrine pancreas) in healthy rat pancreases, which were disrupted in the DCM rat pancreases (deformed boundary and selective β-cell destruction) and restored by PD or metformin treatment. Scale bar = 20 µm.

**Figure 5 cells-13-01196-f005:**
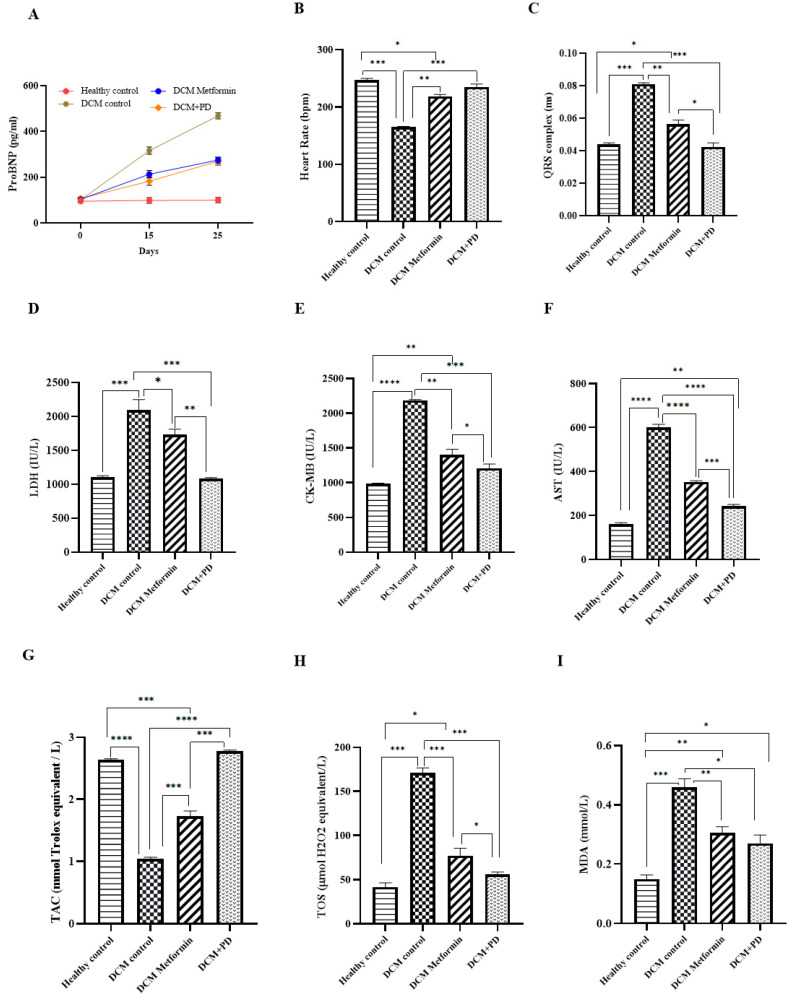
*Phoenix dactylifera* treatment reduced myocardial oxidative stress in the DCM and protects against injury. The DCM rats were treated with or without PD or metformin for 25 days together with the healthy controls prior to sacrifice and collection of myocardial tissue for analysis of (**A**) pro-BNP, (**B**,**C**) ECG, (**D**–**F**) myocardial enzymes, (**G**) total antioxidant capacity, and (**H**,**I**) oxidative stress markers. Data are the means ± SEM, N = 4. **** *p* < 0.0001, *** *p* < 0.001, ** *p* < 0.01, and * *p* < 0.05.

**Figure 6 cells-13-01196-f006:**
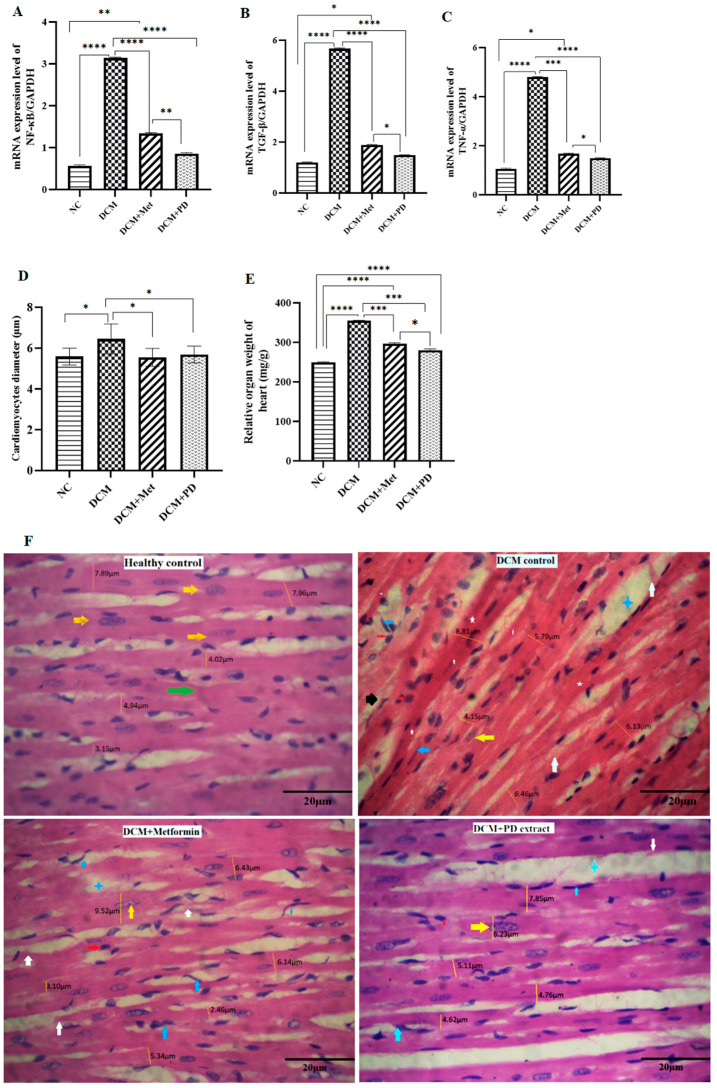
*Phoenix dactylifera* treatment reduces proinflammatory and profibrotic gene expression and restores deranged myocardial structure in DCM. Gene expressions of (**A**) *NF-kB*, (**B**) *TGF-β*, and (**C**) *TNF-α* in the myocardium of healthy control and DCM control rats treated with or without PD or metformin, by RT-qPCR, with normalization to GAPDH as the housekeeping control. (**D**) Cardiomyocyte diameter measured with ImageJ software in a representative histological section of the ventricle (H&E). (**E**) Relative organ weight of heart (mg/g). Data are the means ± SEM, N = 4. **** *p* < 0.0001, *** *p* < 0.001, ** *p* < 0.01, and * *p* < 0.05. (**F**) Representative H&E-stained histopathology sections (40× magnification) showing normal histological features in healthy rat myocardiums characterized by centrally located nuclei in striated fibers (yellow arrows) and uniform intercalated discs (green arrows), which were disrupted in the DCM rat myocardium (focal degeneration, white arrows; edema, blue stars) together with evidence of fibroblasts (blue arrows), fibrosis (white stars), lymphocyte infiltration (red arrows), hemorrhage (black arrows), and cardiomyocyte diameter (yellow straight lines), restored to different degrees by PD or metformin treatment. Scale bar = 20 µm.

**Figure 7 cells-13-01196-f007:**
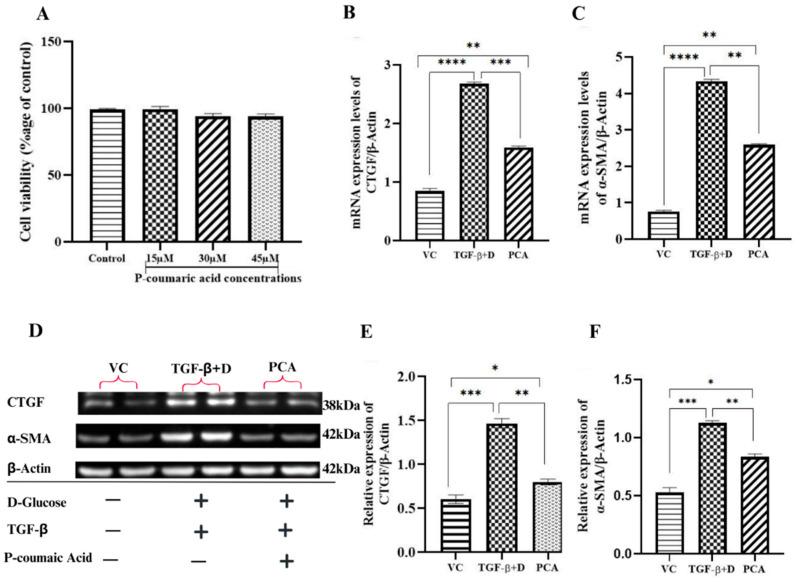
PCA inhibits in vitro myofibroblast differentiation. (**A**) HCF were treated with a range of concentrations of PCA for 3 h prior to analysis of cytotoxicity by MTT assay. HCFs were then stimulated with hyperglycemia and TGF-β with or without PCA compared to vehicle control for 48 h followed by quantifications of CTGF and α-SMA (**B**,**C**) mRNA expressions and (**D**–**F**) protein expression, by qRT-PCR and Western blot, respectively, with normalization to β-actin as the housekeeping control. Data are Mean ± SEM, N = 3 **** *p* < 0.0001, *** *p* < 0.001, ** *p* < 0.01, and * *p* < 0.05.

**Figure 8 cells-13-01196-f008:**
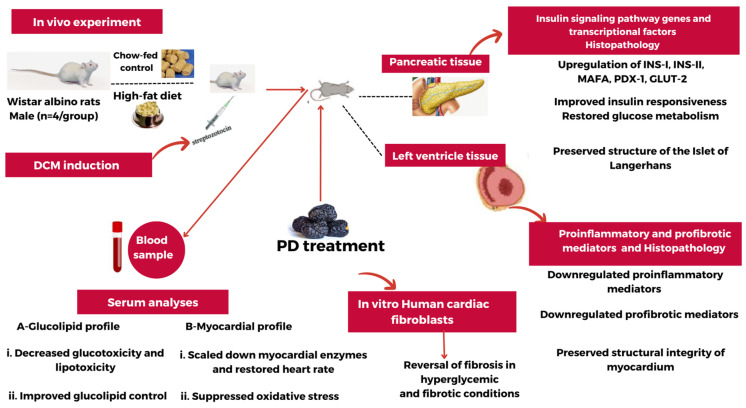
Summary of the overall effects of PD extract on diabetic cardiomyopathy.

**Table 1 cells-13-01196-t001:** Primer sequences for gene expression analysis by qRT-PCR.

Gene	NCBI Gene ID	Nucleotide Sequence	GC%	Annealing Temperature	Product Size
*INS-I* *(Rattus norvegicus*)	16333	FWD: CTGGGAAATGAGGTGGAAAA	45.00	56.65	98
		Rev: TCCACAAGCCACGCTTCTG	57.89	56.89	
*INS-II* *(Rattus norvegicus*)	16334	FWD: AGAAAGGTTTGGTACCTGGAATAGAGC	44.44	53.23	121
		Rev: GTAGAAGAAGCTTCGCTCCCCACA	54.16	53.76	
*PDX-1* *(Rattus norvegicus*)	29535	FWD: CGTAGTAGCGGGACAACGAG	60.00	55.25	71
		Rev: CCCGAGGTTACGGCACAAT	57.89	55.08	
*MAFA* *(Rattus norvegicus*)	366949	FWD: GACCTGATGAAGTTCGAGGTG	52.38	53.39	90
		Rev: GGGCGTCGAGGATAGCGA	66.67	56.29	
*GLUT-2* *(Rattus norvegicus*)	6514	FWD: GGCATGTTTTTCTGTGCCGT	50	59.97	278
		Rev: AAGAACACGTAAGGCCCGAG	55	60.04	
*β-ACTIN* *(Rattus norvegicus*)	60	FWD: CGAGTACAACCTTCTTGCAGC	59.54	52.38	71
		Rev: TATCGTCATCCATGGCGAACTG	60.55	50	
*TGF-β* *(Rattus norvegicus*)	7040	FWD: GGCCAGGAACTACCCCTGTG	65	62.20	254
		Rev: GCCTAGGAGGCATAGAGCGA	60	60.90	
*NF-ĸB* *(Rattus norvegicus*)	4790	FWD: CACCGGATCTTTCCCGCCA	63.16	62.32	155
		Rev: GAGTTTCAGACGCCCGCCTA	60	62.22	
*TNF-α* *(Rattus norvegicus*)	7124	FWD: TGAAGCCCGGCTGATGGTAG	60	61.97	216
		Rev: TGTGGCCATGTCGGTTCACT	55	62	
*GAPDH* *(Rattus norvegicus*)	2597	FWD: TCCTGTTCGACAGTCAGCCG	60	62.14	70
		Rev: CCCCATGGTGTCTGAGCGAT	60	61.98	
*CTGF* (*Homo sapiens*)	1490	FWD: TGGGAGTACGGATGCACTTT	50	59.02	119
		Rev: TACCAATGACAACGCCTC	50	54.63	
*α-SMA* (*Homo sapiens*)	59	FWD: CGTTACTACTGCTGAGCGTGA	52.38	60.14	164
		Rev: AACGTTCATTTCCGATGGTG	45	56.73	
*β-Actin* (*Homo sapiens*)	60	FWD: ACAGAGCCTCGCCTTTGCC	63.16	62.91	71
		Rev: ACAGAGCCTCGCCTTTGCC	63.16	62.91	

**Table 2 cells-13-01196-t002:** The in vitro phytochemical analysis indicates that *Phoenix dactylifera* L. is enriched with flavonoids and phenols. Qualitative analysis indicated detection of key phytochemicals: +++ strongly positive, ++ moderately positive, and + weakly positive. Quantitative phytochemical analysis of total flavonoid content (TFC) and total phenolic content (TPC), mean ± SEM, N = 3.

Phytochemical Analyses		
In Vitro Qualitative Phytochemical Analysis		
Phytochemicals	Test	Detection Level
Carbohydrates	Fehling’s test	++
	Benedict’s test	++
Proteins	Xanthopoietic	++
Alkaloids	Hager’s reagent	+
Flavonoids	NaOH	+
Phenols and tannins	Lead acetate test	+++
Steroids and terpenoids	Chloroform	+
In Vitro Quantitative Phytochemical Analysis		
*Phoenix dactylfera* L.	TFC (mgQE/g)	TPC (mgGAE/g)
	4.82 ± 0.13	58.33 ± 3.24

## Data Availability

Data presented in this study is available on request from the corresponding author.

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
