# Peer review of "Methanolic Extract of Phoenix Dactylifera Confers Protection against Experimental Diabetic Cardiomyopathy through Modulation of Glucolipid Metabolism and Cardiac Remodeling"

_cells, 2024, doi:10.3390/cells13141196_

Round 1

Reviewer 1 Report

Comments and Suggestions for Authors

Nawaz et al. designed a study to explore the cardioprotective potential of Phoenix dactylifera (PD) extract in experimental diabetic cardiomyopathy (DCM). Wistar albino rats were fed high-fat diet (HFD) for 2 months, followed by injection with streptozotocin (STZ; 35mg/kg i.p.) to induce diabetes, then treated with standard therapy, metformin (200mg/kg/day; as reference control), or PD treatment 21 (5mg/kg/day) for 25 days. The authors showed that PD significantly improved fasting blood glucose, insulin and total cholesterol levels and decreased total oxidative stress. Moreover, PD upregulated expression of insulin signaling genes, INS-I and INS-II, and downregulated expression of the pro-fibrotic gene, TGF-β, in pancreas of diabetic rats. Furthermore, PD reduces myocardial oxidative stress (total antioxidant capacity, total oxidative stress and MDA), pro-inflammatory and pro-fibrotic markers and improved cardiac QRS complex widening. Lastly, they demonstrated the antifibrotic effects (reduced CTGF and α-SMA expression) of p-coumaric acid (PCA), a bioactive constituent of PD, in human cardiac fibroblasts under hyperglycemic conditions.

NOVELTY:

The anti-hyperglycemic effects of PD in diabetic rats has been previously established (reviewed in (PMID: 33126433). A previous study (PMID: 29625944) demonstrated the effects of C. flexuosus and P. dactylifera extracts in STZ-diabetic rats. Similar to the current study, they showed improved cardiac markers of oxidative stress (MDA and glutathione peroxidase) and reduced serum markers of cardiac function (creatine phosphokinase-MB). The current study additionally showed the beneficial effects of PD on pancreatic islet insulin-signaling and morphology, supporting previous findings of PD stimulating endogenous insulin secretion from β-cell of pancreatic islets in type I diabetic rats (PMID: 24252889). Whereas the anti-fibrotic effects of PD have been explored in a model of Doxorubicin-induced cardiac dysfunction (PMID: 33804672), but the cardiac benefits of PD in diabetes have been largely unexplored. However, the following suggestions would greatly improve the claim of protection against diabetic cardiomyopathy.

Major comments

1.     Rats were fed high-fat diet (HFD) starting at 2-3 weeks old. This seems very young to start a diet regiment for HFD/STZ-induced diabetes. Please justify choice of model.

2.     The authors defined diabetic cardiomyopathy (DCM) as being characterized by systolic and diastolic dysfunction (line 38-40). Was any functional experimentation performed on the hearts beyond the “cardioprotective index” (Figure 3J), cardiac pro-BNP and ECG (Figure 5A-C)? Echocardiography would be a crucial experiment to address the cardioprotective effects of PD in diabetes by identifying its ability to improve systolic or diastolic dysfunction.

3.     The authors demonstrate reduced TGF expression in hearts of PD-treated diabetic rats. What is the effect of PD on myocardial fibrosis, which is also defined as a key characteristic of diabetic cardiomyopathy. Masson’s Trichrome or Picrosirius Red staining of hearts would greatly add to this finding.

4.     The authors report improvement in pro-BNP, likely as a marker of cardiac function and hypertrophy. However, what was the effect of PD on pathological cardiac hypertrophy? Heart weight and histological determination of cardiomyocyte cell size (WGA) would add to the current finding.

5.     A more thorough discussion of the findings in terms of what has been previously shown in diabetic models is warranted. Ie. PMID: 33126433, 29625944, 24252889

Minor comments

1.     Abstract: Expression of INS-I/II and TGF-β should be clarified in which tissue.

Reviewer 2 Report

Comments and Suggestions for Authors

The authors examined the cardioprotective effects of Phoenix dactylifera (PD) extract in a model of diabetic cardiomyopathy (DCM). Through phytochemical analyses, they used 18 Wistar albino rats, feeding them either a high-fat or standard diet before inducing diabetes with streptozotocin. The rats were divided into four groups: healthy control, DCM control, DCM treated with metformin, and DCM treated with PD extract. After 25 days, they assessed markers of glucolipid balance, oxidative stress, and myocardial health, along with histopathological and gene expression analyses of heart and pancreatic tissues. PD treatment significantly improved glucolipid balance and oxidative stress levels in DCM rats, preserving the structural integrity of the pancreas and heart. Gene expression analysis showed that PD treatment upregulated insulin signaling genes and downregulated profibrotic gene expression compared to the DCM control group. These findings suggest that Phoenix dactylifera may provide cardioprotection in DCM by regulating glucolipid balance and metabolic signaling.

To align aims and conclusions within the abstract and text, it’s crucial to highlight the study’s demonstration of PD extract's potential and propose future research directions. The results section should present data more clearly, avoiding convoluted descriptions. The discussion should interpret the data within the context of existing research, avoiding redundancy by incorporating pertinent studies, such as those identified with the DOIs provided.

doi: 10.3390/molecules25112597.

doi: 10.1371/journal.pone.0296792

doi: 10.1016/j.metabol.2021.154910.

doi: 10.3389/fendo.2023.1106812

doi: 10.1159/000107527.

doi: 10.1186/s40779-023-00506-3.

doi: 10.2337/diabetes.50.9.2105.

The study also needs to acknowledge its limitations and strengths comprehensively, discussing potential biases and their impact on the results. Some conclusions might be overstated and should be moderated, with a stronger emphasis on clinical significance. Including a pictorial or cartoon representation of the main results would enhance the manuscript's overall impact.

Comments on the Quality of English Language

-

Round 2

Reviewer 1 Report

Comments and Suggestions for Authors

The authors claim the current study demonstrates the cardioprotective effect of PD in a rat model of "diabetic cardiomyopathy". However, without echocardiography to measure cardiac systolic and diastolic function, the indirect markers (ie. cardiac pro-BNP) are not sufficient to demonstrate a strong cardioprotective phenotype. Moreover, the addition of cardiomyocyte size would be strongly supported with demonstrated of cardiac remodeling parameters in vivo (LV wall thickness, internal diameter, etc). 

Reviewer 2 Report

Comments and Suggestions for Authors

-

Comments on the Quality of English Language

-

Author Response

No comments to address.